# The Method to Decrease Emissions from Ships in Port Areas

**Vytautas Paulauskas [1,*], Ludmiła Filina-Dawidowicz [2]**  **and Donatas Paulauskas [3]**

[1]   Klaipeda Shipping Research Centre, Klaipeda University, V. Berbomo str. 7-5, LT-92219 Klaipeda, Lithuania

[2]   Faculty of Maritime Technology and Transport, West Pomeranian University of Technology, Szczecin, Ave. Piastów 41, 71-065 Szczecin, Poland; ludmila.filina@zut.edu.pl

[3]   Maritime Engineering Department, Klaipeda University, H. Manto g. 84, LT-92294 Klaipeda, Lithuania; paulauskasd75@gmail.com

\*   Correspondence: vytautaskltc@gmail.com

**Abstract:** Nowadays great attention is being paid to the ecological aspects of maritime transport functioning, including the problem of pollution and emission of poisonous substances from ships. Such emissions have a significant impact on the environment and sustainable operation of ports, especially those located close to intensive waterways. A decrease in emissions from ships may be achieved by implementing different methods, among others, through the use of environmentally friendly fuels, electrical and hybrid vehicles, as well as through the improvement of port approach and inside navigational channels, optimization of the transport processes organization, etc. However, the size of the influence of ships' crew and ports pilots' qualification on the possibility to decrease the emissions from ships during maneuvering in port areas remains a question. This article aims to develop a method to assess the possible decrease of the emissions from ships in ports, considering human factor influence. The method has been developed and verified on the selected case study example. The influence of ships' crew and ports pilots' qualification on time spent on maneuvering operations by ships in port areas and consequently the volume of emissions has been investigated. The research results show that for the set conditions it is possible to reduce emissions from ships up to 12.5%. For that reason, appropriate education and training are needed to improve the qualifications of decision-makers performing ship maneuvers at ports areas.

**Keywords:** maritime transport; emission from ships; sustainable port; energy sources; ship's crew and port pilots qualification; green shipping; environmentally friendly fuels

## 1. Introduction

Maritime transport is extremely important for the world and regional economy; some countries, like those located on islands, are dependent on goods delivered by sea. From 1970 until 2019 the world population increased from 3.7 billion up to over 7.7 billion, but at the same time global maritime trade increased from 2.6 billion tons up to 11 billion tons (in 2018) [1]. In early 2019, the total world fleet constituted 95,402 ships, accounting for 1.97 billion dead-weight tons. Moreover, the carrying capacity grew by 2.61%, compared with the beginning of 2018 (Table 1). It is forecasted that ship number and size will continue to grow [1].

Nowadays, much attention is paid to the issues of the negative impact of shipping on the environment. It is assumed that maritime transport emits around 940 million tons of $CO_2$ annually, being responsible for about 2.5% of global greenhouse gas (GHG) emissions [2–4]. The negative impact of shipping on the environment is concentrated essentially in selected areas: close to big ports (e.g., Shanghai, Rotterdam, Hamburg, and others), main waterways (e.g., Suez and Panama channels),

approach channels or rivers (e.g., Elbe river providing the Hamburg port, Yangtze river–to Shanghai port), and other areas [5–9].

**Table 1.** World fleet by principal vessel type 2018–2019 (thousand dead-weight tons) [1].

| Principal Types | 2018 | 2019 | Percentage Change 2019/2018 |
|---|---|---|---|
| Oil tankers | 562,035 | 567,533 | 0.98 |
| Bulk carriers | 818,921 | 842,438 | 2.87 |
| General cargo ships | 73,951 | 74,000 | 0.07 |
| Container ships | 253,275 | 265,668 | 4.89 |
| Gas carriers | 64,407 | 69,078 | 7.25 |
| Chemical tankers | 44,457 | 46,297 | 4.14 |
| Offshore vessels | 78,269 | 80,453 | 2.79 |
| Ferries and passenger ships | 6922 | 7097 | 2.53 |
| Other/not available | 23,946 | 23,929 | -0.07 |
| World total | 1,926,183 | 1,976,491 | 2.61 |

The International Maritime Organization (IMO) constantly pays a lot of attention to the issues of reduction of emissions from ships, especially decrease of sulfur ($SO_x$), nitrogen ($NO_x$), carbon dioxide ($CO_2$), particulate matter (PM), and other substances [2,4]. In 2008 the Sulfur Emission Control Areas (SECA) in the Baltic Sea started to be created, then they covered the North Sea and the English Channel. In 2012 similar Emission Control Areas (ECA) were created in the North American area, and the decision to decrease $SO_x$ in ships fuel was taken globally [4]. Ships' fuel sulfur requirements worldwide and in the ECA are presented in Figure 1.

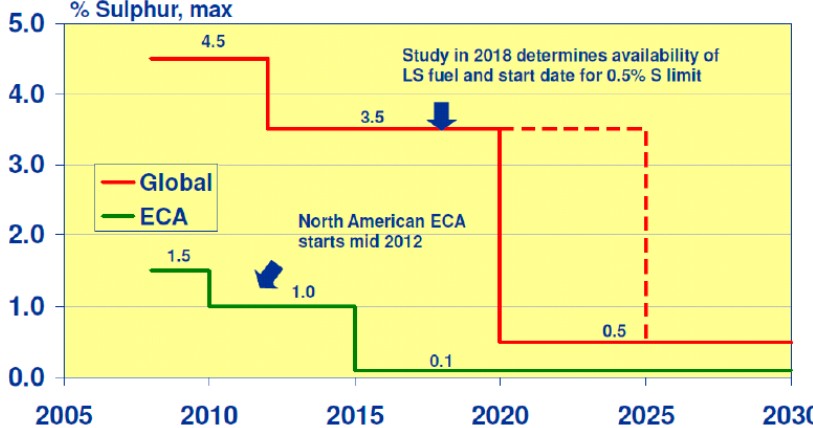

**Figure 1.** Ships' fuel sulfur requirements worldwide and in Emission Control Areas (ECA) [10].

This problem is recognized globally. For example, according to the European Union (EU) decision, the emission from ships should be decreased in 2020 not less than 20% (in comparison to 2005) and in 2030 not less than 30%. Moreover, it is estimated that daily suboptimal operation of ships increases energy consumption up to 15%–18% caused by inappropriate decisions by the ship's crew, which influence the increase of emission from ships accordingly [1,11].

The conducted analysis of available literature revealed that green shipping development and environmental sustainability in seaports are frequently discussed research topics. The reviewed studies analyze technical and technological aspects of sustainable shipping, show organizational challenges and possible economic effects, assess the volume of pollution, and propose ways to decrease it [12–17].

Different regulations and approaches are implemented to reduce the volume of emissions. These approaches include, i.a., the decrease of emissions from vehicles using environmentally friendly fuels, optimization of ships' maneuvers in port approaches and internal navigational channels, by optimal design of the ports' terminals, as well as minimization of vehicle service time, improvement of ports'

connection with hinterland, and others [18–29]. The implementation of a large number of these approaches requires significant financial outlays that may deal with investments in the development of ports' infrastructure, construction changes of ships, etc.

Conducted research studies also pay attention to human factor influence on navigation safety and ship operation [30–33]. However, it should be noted that the available literature positions do not present a detailed analysis of the influence of ships' crew and ports pilots' qualification on the possibility to decrease the emissions from ships during ships' maneuvering in seaports.

In this article the human factor influence on emissions from ships in port areas is analyzed in detail. It aims to develop a method allowing an assessment of the possible decrease in the emissions from ships, considering ships masters' and ports pilots' qualification. The research questions were formulated as follows:

1. Are the emissions from ships in seaports influenced by ships masters' and ports pilots' qualification?
2. What is the volume of emissions from ships that may be reduced during ships' maneuvering operations in port area depending on responsible person's qualification?

It is assumed that in case the maneuvers of ships will be done by operators with different qualifications, there will be differences in the volume of emissions created by ships. The proposed method is based on empirical data analysis and shows the way to analyze the data using dispersion and maximal dispersion methods. Its idea is to show that emissions from ships in ports may be reduced by employing ships' masters and ports' pilots with appropriate qualifications. The case study analysis is used to verify the method. Real data of ship sailing in seaports are considered, bearing in mind that ships' maneuvers are performed by different operators. Identified differences in ship sailing parameters allow an estimation of the emissions created by ships and share of possible emission reduction.

Section 2 of the paper presents the literature analysis. Section 3 describes the methodology used to conduct the research. The results of case study analysis are shown in Section 4. The paper is summarized by discussions, conclusions, and directions of future research, presented in Sections 5 and 6.

## 2. Literature Analysis

Emissions of poisonous substances are produced by different sectors of the economy and different approaches to estimate the volume of emissions are implemented [5,34–38]. This also applies to transport and logistics systems functioning in pursuit of sustainable development of these systems [39–43]. Direct and indirect gas emissions caused by water transport are noted, especially air pollution through various greenhouse gas emissions ($SO_2$, $NO_x$, $CO_2$, $PM_{2.5}$, $PM_{10}$) [32,44–46].

Emission from ships has wide regulation framework. Stringent regulations were implemented by the IMO and the EU (i.e., within and beyond the SECA limits) to reduce the sulfur emissions of ships [4,10,45]. IMO has tasked its members to achieve a 70% reduction in $CO_2$ emissions by 2050, the decarbonization process is based on EU strategic documents, low-emission and zero-emission technologies [12]. Furthermore, the nitrogen emission control area (NECA) in the Baltic Sea and the North Sea is planned to be introduced in 2021 [15]. These and other restrictions significantly affect the functioning of shipping companies and seaports, as well as forces them into action measures to comply with environmental protection requirements [47].

Bouman et al. provided a comprehensive overview of the $CO_2$ emission reduction potentials and measures and stated that emissions can be reduced by more than 75%, based on current technologies and by 2050, through a combination of measures if policies and regulations are focused on achieving these reductions [48]. Available studies recognize the need for research result implications for the further development of policies addressing sustainability in shipping management [49], as well as combining instruments into policy packages, and emphasise the urgency of addressing technology and policy solutions for the maritime sector [50]. It is highlighted that management policies should depend

on the internal and external environment of shipping companies [13]. Safe, secure, energy-efficient, affordable, reliable, climate-resilient, low-carbon, and rule-based maritime transport systems contribute to achieving an economically efficient and environmentally sound development [14,51].

Decreasing of emissions from shipping is the multipurpose task which links technical, technological, organizational, legal decisions, as well as human factor influence (employee education and training). In this regard, different approaches may be found in the literature [52]. In order to reduce emission volume in port areas and main international waterways, environmentally friendly fuels and renewable energy resources may be used [2,3,19,20,22,24,28,37,53]. In many cases, ships in ports use only permitted fuels (depending on fuel content). Improvements in ship structure, including used engines, are also investigated [3,42,54–56]. Slow steaming strategies have been introduced in most shipping lines and significantly decrease $CO_2$ emissions from international shipping [16]. Moreover, simplified and composite ship fuel consumption models for ocean-going vessels were developed and facilitate the assessment of the fuel volume needed for ships [32,46,57–59]. It is also noted that modal shift policy is one of the ways to reduce emissions and should take into consideration environmental strategy and possible pollution reduction [60].

The analysis of the literature positions revealed that currently problems of decreasing the emissions from ships are applied to different shipping areas, however a lot of attention is paid to ships' service in seaports. Ports, as the important nodes of transport and logistics systems, make an effort to plan their territory optimally; however, it is not an easy task, especially when ports are located in cities or next to them [8,61–63]. It was estimated that in some specific harbor areas in Asia, ships can contribute up to 7%–26% to the local fine particulate matter concentrations [64]. Attention is paid to the need to develop approaches for green ports that have emerged within environmental management and give attention to the ecological issues [17]. This reinforces the belief that actions should be continued to reduce the volume of emissions in ports.

The complicated design of navigational approach and inside port channels sometimes need a lot of ships maneuvers that may be optimized [65,66]. Emissions from ships, especially observed in approaches to the ports and in ports areas, is influenced by the performed maneuvers and often requires changing the ship's engine power, which should be carried out by the ship's crew and port pilot with appropriate qualifications.

A comprehensive review of ships' maneuvers and environmental effects was conducted by Di Vaio et al. [67]. It was mentioned that in order to achieve high competitiveness in seaports in response to environmental and energy regulation, port authorities, users, and local communities have to invest considerable resources. The paper introduces managerial key performance indicators to support port authorities in their decision-making processes, considering inter-organizational relationships with shipping lines that aim to develop environmentally sustainable and energy efficient ports.

It should be noted that port configuration and ship maneuvering areas are different in particular ports [26,66]. Two main factors influence the emissions produced by ships in ports: types of maneuver operations made by ships and efficiency of tug assistance. On the one hand, ships sailing in port areas have to be safe. On the other hand, it is very important to optimize the time of ships' movement and minimize maneuvers inside the port that mainly depend on the responsible people's qualifications and health [26,29,68]. The reviewed research papers mention that these qualifications are influenced by the region seafarers come from [69] and affect the safety in maritime operations [30,70]. However, the influence of ships' crew and port pilots' qualification on the possibility to reduce the emissions from ships has not been analyzed in detail.

Corrigan et al. [31] stated that there is an increasing awareness of human factor influence in maritime transport and much more focused research is required concentrating on the specific complexities, constraints, and shared processes of port environments. It is also highlighted that maritime educational institutions around the world should be prepared to provide the skilled labor the industry will require to remain competitive [33].

On the basis of the conducted literature analysis it should be stated that:

- the problem of decreasing the emissions from ships is up-to-date and further solutions in this field should be developed;
- there is a need to look for solutions to reduce the emissions from ships that will not require high volumes of investments;
- human factor influence on poisonous substance emissions from ships has been analyzed so far to a small extent.

This justifies the need to investigate the impact of ships' crew and port pilots' qualifications and decisions on ships' maneuver operations in port areas, as well as further search for ways to decrease emissions from ships.

## 3. Materials and Methods

The following methodology was used to develop the method aiming to assess the possible decrease of the emissions from ships (Figure 2). After the literature analysis the necessary data were collected to develop the method. The method was verified on the basis of Klaipeda port case study analysis. Then, the appropriate conclusions were drawn.

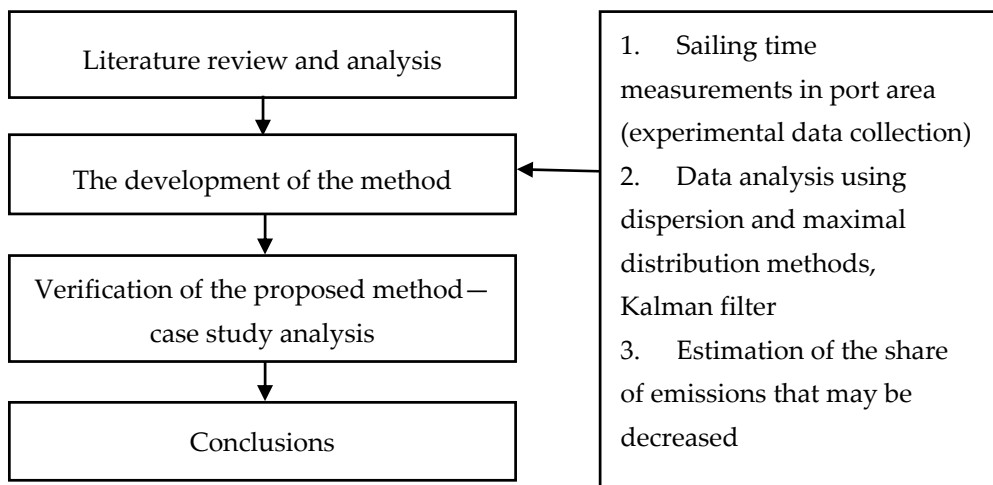

**Figure 2.** The methodology used to conduct the research.

To decrease the energy consumption, as well emissions from shipping activity, it is necessary to clearly understand ships' operation processes and have knowledge about emission sources, including transport means and port equipment operation, possibility to use the definite fuels for the different machines' functioning, as well as to know the methods of optimal holistic port and terminal design, and recognize the significance of the high qualifications of port staff and ships' crew.

Emissions from ships and other transport vehicles directly depend on the quantity and quality of fuel used. The main emissions from ships constitute: carbon dioxide ($CO_2$), nitrogen oxides ($NO_x$), carbon monoxide (CO), sulfur oxides ($SO_x$), and particulate matter (PM) [10].

Carbon dioxide and sulfur oxide emissions could be calculated as follows (Equations (1) and (2)):

$$CO_2 = k_{CO_2} \cdot Q_f, \tag{1}$$

$$SO_x = k_{SO} \cdot Q_f, \tag{2}$$

where: $k_{CO_2}$ is the carbon dioxide coefficient, which depends on fuel quality (for the high-quality diesel fuel this coefficient is between 3.0 and 3.2 and for the LNG (Liquefied Natural Gas) fuel this coefficient is between 2.1 and 2.3); $k_{SO}$ is the sulfur oxides coefficient, which depends on the fuel quality (sulfur

percentage in fuel), for example, for $SO_x$ control areas it is 0.001 for diesel fuel and 0.0 for the LNG fuel; $Q_f$ is the quantity of fuel during ship sailing, which can be estimated as follows (Equation (3)):

$$Q_f = \int_0^T k_f \cdot q_f \cdot N_{av} \cdot dt, \tag{3}$$

where $k_f$ is the coefficient, which depends on the type of engine [10,71]; $q_f$ is the consumption of fuel for the definite engine (kg/kWh), $N_{av}$ is the ship's engine's average power during the sailing period (kW), which can be calculated using Equation (4):

$$N_{av} = \frac{\int_0^t N_i \cdot dt}{t}, \tag{4}$$

where $N_i$ is the instantaneous ship's main engine power (kW); $t$ is the ship's sailing time.

Nitrogen oxides, carbon monoxide, and particulate matter emissions depend on the real engine power, type of fuel, and its quality, and can be assessed as follows (Equations (5)–(7)):

$$CO = k_{CO} \cdot N_{av} \cdot t, \tag{5}$$

$$NO_x = k_{NO_x} \cdot N_{av} \cdot t, \tag{6}$$

$$PM = k_{PM} \cdot N_{av} \cdot t, \tag{7}$$

where $k_{CO}$ is the carbon monoxide coefficient, which depends on fuel quality and engine condition, for the high-quality diesel fuel and modern engines it varies between 0.005 kg/kWh for diesel fuel and 0.003 kg/kWh for LNG fuel; $k_{NO_x}$ is the nitrogen oxides coefficient, which also depends on fuel quality and engine state, for the high-quality diesel fuel and modern engines it is in the range of 0.008–0.0012 kg/kWh for diesel oil and 0.003–0.004 kg/kWh for LNG fuel; $k_{PM}$ is the particulate matter coefficient, which depends on fuel quality and engine, for the high-quality diesel fuel and modern engines it varies between 0.0005–0.0006 kg/kWh for diesel oil and about 0.0001 kg/kWh for LNG fuel [10,72].

As it was mentioned before, the staff (ships' crew and ports pilots') qualifications and behavior greatly influence the volume of emissions coming from ships. The time the ship spends on crossing the port area and maneuvering operations affects the fuel consumption and consequently the volume of produced emissions. In order to evaluate this influence, it is proposed to use experimental data and apply dispersion and maximal distribution methods for their analysis.

The developed method is focused on time and emission bands analysis. In order to calculate the size of random error or time bands, we use dispersion and/or "maximal distribution" mathematical methods. It was set that the size of random error ($e$ or $\Delta t_P$) in the dispersion method is comparable with dispersion ($\sigma_y$) [26,73,74]. Dispersion method implementation to evaluate the ship's sailing time in port time bands can be expressed using Equation (8) [73,75]:

$$\sigma_y^2 = \frac{1}{n-1} \sum \left( t_i - t_y \right)^2, \tag{8}$$

where $n$ is the number of the measurements; $t_i$ is the particular measurement results (ship's sailing time in port area); $t_y$ is the mathematical expectation of the average sailing time, which can be calculated as follows (Equation (9)):

$$t_y = \frac{\sum_{i=1}^n t_i}{n}. \tag{9}$$

Finally, sailing time band with determined probability (e.g., 63%–68%) ($\Delta t_P$) can be presented by random error (Equation (10)):

$$e = \Delta t_P = \pm \sqrt{\sigma_y^2}. \tag{10}$$

Sailing time band ($t_P$) may be calculated using the Equation (11):

$$t_P = t_y \pm \Delta t_P. \tag{11}$$

Similarly, sailing time band can be estimated using the "maximal distribution" method. For the purpose of the research it can be expressed as follows (Equation (12)) [26,76]:

$$t_P = t_y \pm P' \cdot \Delta t \cdot k_t, \tag{12}$$

where $P'$ is the probability coefficient (it has been proposed that in the case of probability 63%–68%, the coefficient should equal 1, in the case of probability 95%, the probability coefficient should be 2, and in the case of probability 99.7%, the probability coefficient equals 3); $\Delta t$ is the difference between sailing times; $k_t$ is the coefficient, which depends on the number of measurements (the number of possessed data): in the case the number of data is 3, this coefficient will be 0.55; in the case of data numbering 4, this coefficient will be 0.47, and similarly depending on the number of obtained data 5—0.43; 6—0.395; 7—0.37; 8—0.351; 9—0.337; 10—0.329; 11—0.325; 12—0.322 and so on, but the minimum value of this coefficient could be about 0.315, in the case the number of collected data will be more than 15.

The proposed approach introduces a new way to calculate and analyze ships' sailing time. In particular, it deals with sailing time band (most probable interval) calculation on the basis of experimental (real) data using request filtration to receive sailing time calculated values that are as close as possible to the actual ship's sailing time. The probability of the ship's sailing time or differences between average and actual sailing time data are the stochastic processes, therefore, filtration is required. To evaluate the difference between expected and real data, we implement the Kalman filter (other filters also may be used), which may be assessed as follows (Equations (13) and (14)) [77]:

$$x_k = Ax_{k-1} + Bu_k + \omega_k, \tag{13}$$

with observations $z_k$:

$$z_k = Hx_k + v_k, \tag{14}$$

where $A$, $B$, $H$ are coefficients; $\omega_k$, $v_k$ are the sequence of noisy observations; $x_k$, $u_k$ are control vectors.

The same approach is implemented to analyze the emission bands for the particular substances. Comparing the experimental results preceded by dispersion and maximal distribution methods, it is possible to evaluate the share of emissions that may be decreased. It should be highlighted that the presented methodology may be implemented to analyze the emissions from any kind of ship operating in any port.

## 4. Results of Case Study Analysis

In order to verify the proposed method, the case study was considered. Klaipeda port was selected and LNG standard tankers' movement through the port area from approach channel up to ships' turning basin was examined. The vessel was navigated by the ship's master and port pilot. Additionally, the behavior of 10 operators, who had professional experience to navigate ships using simulators, was investigated. These operators took part in the experiment and used the "SimFlex Navigator" simulator to steer a similar ship on the selected route.

While investigating the emissions from ships sailing through the port area, the external forces caused by wind, current, waves, and shallow water effect, influencing the ship's maneuverability, were also taken into account. The experiment was conducted under the set conditions, typical for the big ships entering Klaipeda and other ports.

The experimental study was carried out in Klaipeda port in winter 2019 when an LNG Standard tanker (with a capacity about 150,000 m$^3$ of LNG) entered the port and sailed to the LNG terminal (Figure 3). The ship's movement parameters as engine power and ship's speed were recorded by the ship's equipment; additionally, the ship's speed was measured by differential GPS and checked by AIS

(Automatic Identification System). The sections of the vessel's sailing route are presented in Figure 4. The sailing time was measured in particular measurement points.

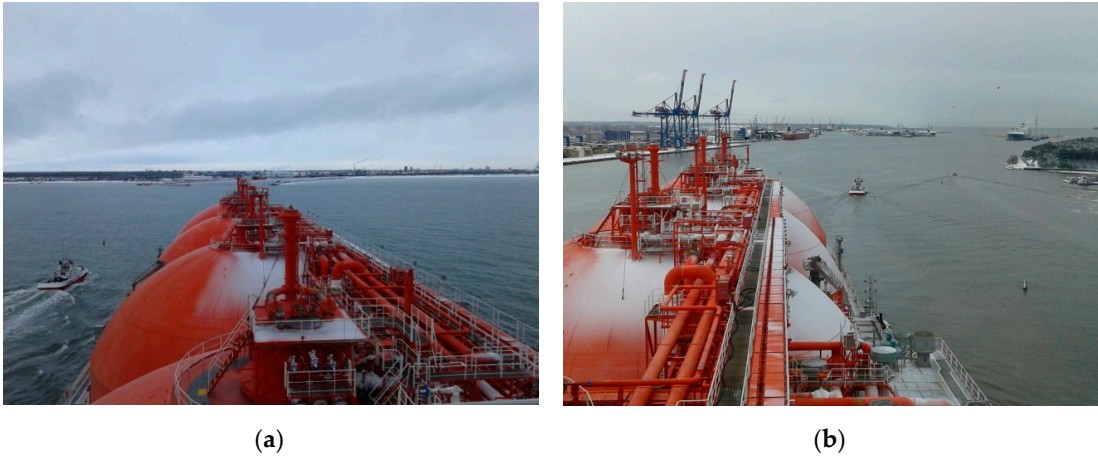

(**a**)　　　　　　　　　　　　　　(**b**)

**Figure 3.** LNG Standard tanker: (**a**) measurement start position; (**b**) tanker passing the navigational channel of Klaipeda port.

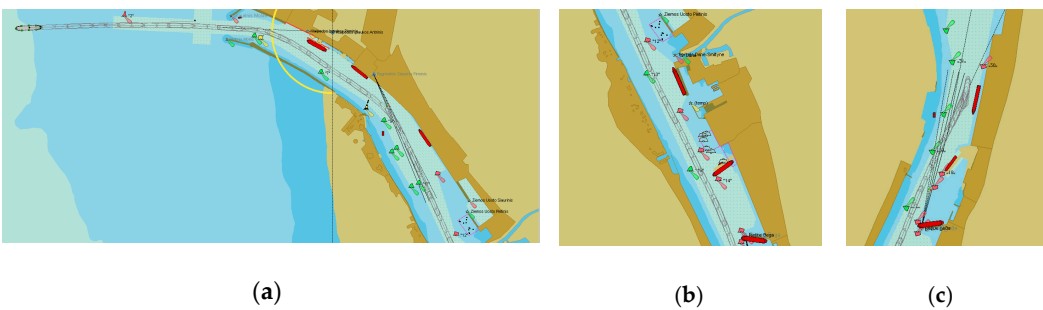

(**a**)　　　　　　　　　　(**b**)　　　　　　　　(**c**)

**Figure 4.** LNG Standard tanker sailing route: (**a**–**c**) route sections in the navigation channel of Klaipeda port.

The calculations for the specific case study were conducted under the set assumptions. The LNG Standard tanker that passed through the port area had the following parameters: length ($L$)—290 m, width ($B_1$)—49 m, draft ($T_1$)—12 m, displacement ($D_1$)—125,000 t. Additionally, it was set that the block coefficient ($\delta$) was 0.75, ship's speed ($v_1$) varied between 6 and 9 knots (from 3.1 up to 4.6 m/s), depth in port area ($H$) was 14.5 m. The LNG Standard tanker main sailing parameters, measured during the experiment performed by selected operators, are shown in Figure 5.

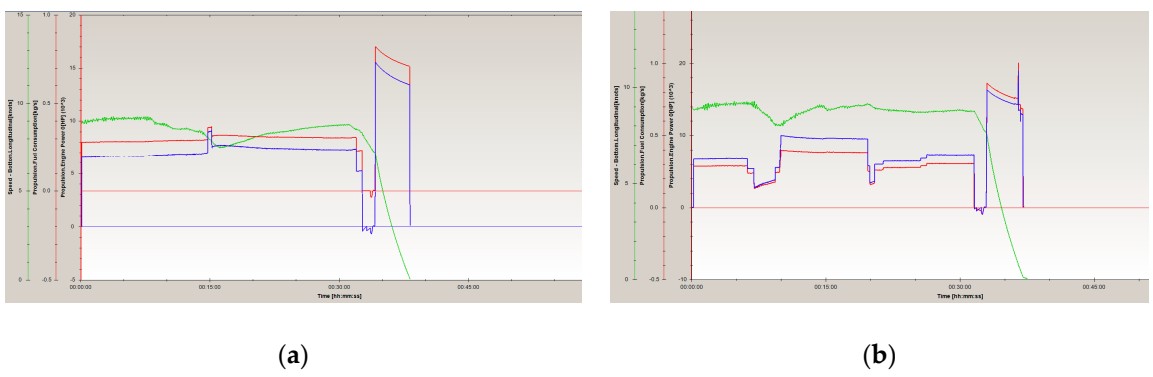

(**a**)　　　　　　　　　　　　　　(**b**)

**Figure 5.** The LNG Standard tanker main sailing parameters ($v_i$, $Q_{fi}$, $N_i$ as function of $t$), received during conducting the experiment: (**a**) ship steered by operator No. 4; (**b**) ship steered by operator No. 7.

The settings used in the simulator were adjusted to external conditions measured during the performance of experimental study on the real ship. These settings included: wind direction SW, velocity 10 m/s, current out of port 0.5 knots, waves SW direction, height 1 m in approach channel and 0 inside port, shallow water effect ratio (ship's draft to depth) T/H = 0.82. In total, over 100 similar experiments were performed using the simulator, among which 10 experiments on a real ship under similar external conditions (wind, current, waves, shallow effects) were carried out by different operators. By keeping identical sailing conditions for ships' maneuvers, it was possible to repeat the experiment and identify the behavior of different operators.

On the basis of the conducted measurements it was possible to compare and analyze the achieved results. The analysis covered experimental data received during the observation of 10 operators' behavior and real ship sailing. The selected investigated parameters after filtration by Kalman filter are presented in Table 2.

**Table 2.** Selected experimental data gained for LNG Standard tanker sailing in Klaipeda port, obtained from the "SimFlex Navigator" simulator and real ship.

| Operator | Sailing Time, $t_i$, min | Average Engine Power, $N_{av}$, kW | Used Fuel, $Q_f$, kg |
|---|---|---|---|
| 1 | 37.5 | 6460 | 910 |
| 2 | 38.2 | 6410 | 960 |
| 3 | 36.9 | 6520 | 1020 |
| 4 | 38.5 | 6420 | 980 |
| 5 | 37.8 | 6460 | 950 |
| 6 | 37.0 | 6610 | 1015 |
| 7 | 39.2 | 6380 | 925 |
| 8 | 38.1 | 6220 | 890 |
| 9 | 36.7 | 6610 | 1050 |
| 10 | 38.2 | 6510 | 990 |
| Real Ship | 37.5 | 6220 | 880 |

On the basis of the received experimental results the emissions in the case of using the diesel oil with 0.1% $SO_x$ content and LNG fuel (real LNG Standard tanker used LNG fuel) were calculated. Calculation results achieved using the methodology presented in the article are shown in Table 3.

**Table 3.** Emissions from LNG Standard tanker during sailing in Klaipeda port calculated for diesel oil/LNG usage, based on experimental results obtained from the "SimFlex Navigator" simulator and real ship.

| Operator, No. | $CO_2$, kg | $SO_x$, kg | CO, kg | NO, kg | PM, kg |
|---|---|---|---|---|---|
| 1 | 2821/2002 | 0.91/0 | 32.3/22.6 | 6.5/4.5 | 3.23/0.32 |
| 2 | 2976/2112 | 0.96/0 | 32.1/22.4 | 6.4/4.5 | 3.21/0.32 |
| 3 | 3162/2244 | 1.02/0 | 32.6/22.8 | 6.5/4.6 | 3.26/0.33 |
| 4 | 3038/2156 | 0.98/0 | 32.1/22.5 | 6.4/4.5 | 3.21/0.32 |
| 5 | 2945/2090 | 0.95/0 | 32.3/22.6 | 6.5/4.5 | 3.23/0.32 |
| 6 | 3146/2233 | 1.02/0 | 33.1/23.1 | 6.6/4.6 | 3.31/0.33 |
| 7 | 2868/2035 | 0.93/0 | 31.9/22.3 | 6.4/4.5 | 3.19/0.32 |
| 8 | 2759/1958 | 0.89/0 | 31.1/21.8 | 6.2/4.4 | 3.11/0.31 |
| 9 | 3255/2310 | 1.05/0 | 33.05/23.1 | 6.6/4.6 | 3.31/0.33 |
| 10 | 3069/2178 | 0.99/0 | 32.6/22.8 | 6.5/4.6 | 3.26/0.33 |
| Real LNG tanker | 2759/1958 | 0.89/0 | 32.1/22.5 | 6.4/4.5 | 3.21/0.32 |

On the basis of the received experimental data, the mathematical expectation and bands (time and emission) for the specific ship operation were calculated using methods presented in the article. The achieved calculation results are presented in Tables 4 and 5. Dispersion and maximal distribution methods were used to assess the main ship's sailing and emission parameters.

**Table 4.** Main ship's operation parameters and bands.

| Parameter | Mathematical Expectation | Parameter's Band Received by Dispersion Method | Parameter's Band Received by Maximal Distribution Method |
|---|---|---|---|
| Sailing time, min | 37.9 | 37.1–38.7 | 37.1–38.7 |
| Engine power, kW | 6456 | 6346–6566 | 6329–6583 |
| Fuel consumption, kg | 962 | 908–1016 | 910–1014 |
| $CO_2$ (diesel fuel), kg | 2991 | 2822–3160 | 2830–3152 |
| $CO_2$ (LNG fuel), kg | 2116 | 1996–2236 | 2002–2230 |
| $CO$ (diesel fuel), kg | 32.3 | 31.8–32.9 | 31.7–32.9 |
| $CO$ (LNG fuel), kg | 22.6 | 22.2–23.0 | 22.2–23.0 |
| $NO_x$ (diesel fuel), kg | 6.5 | 6.37–6.63 | 6.37–6.63 |
| $NO_x$ (LNG fuel), kg | 4.5 | 4.41–4.59 | 4.43–4.57 |
| $PM$ (diesel fuel), kg | 3.23 | 3.17–3.29 | 3.16–3.30 |
| $PM$ (LNG fuel), kg | 0.32 | 0.30–0.34 | 0.31–0.33 |
| $SO_x$ (diesel fuel), kg | 0.96 | 0.90–1.02 | 0.91–1.01 |
| $SO_x$ (LNG fuel), kg | 0 | 0 | 0 |

**Table 5.** The bands of the selected ship's sailing and emission parameters received by dispersion and maximal distribution methods.

| Ship's Sailing and Emission Parameters | Parameter's Band Received by Dispersion Method, % | Parameter's Band Received by Maximal Distribution Method, % |
|---|---|---|
| Sailing time | 4.2 | 4.2 |
| Engine power | 3.4 | 3.9 |
| Fuel consumption | 11.2 | 10.8 |
| $CO_2$ (diesel fuel) | 11.3 | 10.8 |
| $CO_2$ (LNG fuel) | 11.3 | 10.8 |
| $CO$ (diesel fuel) | 3.4 | 3.7 |
| $CO$ (LNG fuel) | 3.5 | 3.5 |
| $NO_x$ (diesel fuel) | 4.0 | 4.0 |
| $NO_x$ (LNG fuel) | 4.0 | 3.1 |
| $PM$ (diesel fuel) | 3.7 | 4.3 |
| $PM$ (LNG fuel) | 6.5 | 6.2 |
| $SO_x$ (diesel fuel) | 12.5 | 10.4 |
| $SO_x$ (LNG fuel) | 0 | 0 |

The achieved research results (Table 4) show that both methods for the band calculation (dispersion and maximal distribution methods) may be implemented for the analysis of ships' sailing and emission parameters. The difference between the results obtained using these two methods is less than 1%. At the same time, the differences in particular operators' qualifications and behaviors could be observed. This demonstrates that decisions taken by operators significantly influence the ship's sailing time, affect ships' fuel consumption, and consequently volume of emission (Table 5).

It should be also mentioned that fuel type influences the emission volume. The research results made it possible to compare the emissions coming from the operation of ships using diesel and LNG fuels. Usage of fuels, like LNG, can decrease the volume of emissions from ships up to 30% and $SO_x$ emission up to 100%. This underlines the effectiveness of their use in emission reduction.

## 5. Discussion

The research results reveal that qualifications of ports' pilots and ships' masters play a significant role in performing ships maneuvers, and consequently the volume of emissions from ships.

The number of conducted measurements during the case study analysis can be discussed. This number was limited, however, representative for the established research topic. The differences in operators' behavior while performing maneuvering operations were visible and proved that the level of pilots' qualification was different. Therefore, it should be stated that research results are satisfactory and allowed to answer the first research question—the emission from ships in seaports is influenced

by ships masters' and port pilots' qualification. In our future research we will try to extend the number of measurements and involve more qualified pilots, who will agree to take part in the study.

Conducted experimental results showed that the volume of emissions from ships may be reduced by 12.5% or even more. Some literature sources mention that the way of maneuvering the ship may influence the emission volume up to 15%–18% [72]. It should be noted that the research presented in this article was conducted for the specific ship and defined sailing conditions limited to the port area, that also influenced the results. However, it was possible to answer the second research question and assess the volume of emission from ships that may be reduced during ships' maneuvering operations in port area depending on responsible person qualification.

It should be highlighted that the research results were also influenced by external conditions that were founded during the study. Therefore, it will be reasonable to repeat experiment considering external conditions in different seasons and compare the results. On that basis, it will be possible to define the external conditions during which it is particularly important to have high qualifications of staff to make the right decisions and reduce the volume of emission.

Moreover, research results may have managerial implications. Seaports, as well as shipping companies, may change their procedures and introduce strict conditions of skill verification during employee hiring and professional work, in pursuit of reducing the volume of emissions at seaports. Companies may organize regular trainings and invest in employee education aiming at improving staff qualifications in supporting decision-making during maneuver operations. These activities may affect the development of companies' environmental policy in order to decrease the costs of ship operation, as well as emission volume.

The achieved results also proved that the maritime education quality is very important to obtain the necessary qualifications for ships operators. This justifies the need to raise the quality of professional education at the universities and increase the number of practical hours on simulators for seafarers, which will enable an increase in their qualifications and attractiveness on the labor market.

## 6. Conclusions

In shipping areas like ports, approach channels, and main waterways (Panama, Suez channels etc.), where intensive traffic is observed, the decrease of emissions from ships is very important for the sustainable development of ports. It influences not only people's health and quality of life, but also the surrounding environment.

The study presented in this article, aimed to develop a method to assess the possible decrease of emissions from ships, considering ships operators' qualification. This goal has been achieved. Study results show that qualification of the ship's masters and port pilots can influence the emission volume. This volume may be decreased up to 12.5% or even more.

Achieved results also allowed a comparison of the emissions coming from the operation of ships using diesel and LNG fuels. It was stated that the usage of more environmentally friendly fuels, like LNG, can decrease emissions from ships up to 30% and $SO_x$ emission up to 100%.

The developed method presents a way to analyze empirical data and may be introduced in practice. It shows the role of operators' education and training, as well as justifies the need for regular improvement of staff qualifications. Moreover, the presented approach may be useful for seaports and shipping companies and may be implemented to assess the personal qualifications during the selection of staff responsible for ships' steering.

More detailed and complex investigations of external factors influencing the volume of emission coming from ships in port areas, like the type of tugs used, wind, and currents, will form the direction of our further research.

**Author Contributions:** Conceptualization, V.P. and D.P.; methodology, V.P.; software, V.P.; validation, V.P. and D.P; formal analysis, L.F.-D.; investigation, V.P., L.F.-D. and D.P.; resources, V.P. and D.P.; data curation, V.P. and D.P.; writing—original draft preparation, V.P. and D.P.; writing—review and editing, L.F.-D.; visualization, V.P.,

L.F.-D. and D.P.; supervision, V.P.; project administration, V.P.; funding acquisition, V.P. All authors have read and agreed to the published version of the manuscript.

**Funding:** This research received no external funding.

**Acknowledgments:** This article is based on the research conducted within the Interreg South Baltic project SB Transport LOOPS co-financed by the European Union from the European Regional Development Fund.

**Conflicts of Interest:** The authors declare no conflict of interest.

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
