# Peer review of "The Method to Decrease Emissions from Ships in Port Areas"

_sustainability, doi:10.3390/su12114374_

Round 1
Reviewer 1 Report
I read the paper and these are my recommendations for authors:
- The second chapter look also like a Introduction. The title does not correlate with the content
- Please explain what is circular error. Why is equal with dispersion?
- In chapter 3 equation editor settings are not good
- Number of chapter Results is 4 not 3
- For chapter 4 is more appropriate a title as a Case Study
- The number of of measurements can be higher. It is not clear how the separation of external factors was achieved in measurements (wind, waves, etc.)
Author Response
Suggestion 1: The second chapter look also like a Introduction. The title does not correlate with the content
Respond: The second chapter was rewritten. More issues related to literature analysis were added. The chapter was titled “Literature review”. The number of reviewed literature positions was extended up to 77. The analysis of these positions allowed positioning the research appropriately.
Suggestion 2: Please explain what is circular error. Why is equal with dispersion?
Respond: Circular error is used in navigation. In this Article changed in to random error.
Suggestion 3: In chapter 3 equation editor settings are not good
Respond: The equations in Section 3 were rewritten using another equations editor.
Suggestion 4: Number of chapter Results is 4 not 3
Respond: The chapter’s number have been changed.
Suggestion 5: For chapter 4 is more appropriate a title as a Case Study
Respond: The chapter’s title was changed into “Results of case study analysis”.
Suggestion 6: The number of measurements can be higher. It is not clear how the separation of external factors was achieved in measurements (wind, waves, etc.)
Respond: In the research 11 measurements were conducted (one measurement was done on real ship example and 10 measurements were conducted on simulator in the laboratory, when ships pilots performed manoeuvring operations). The total number of measurements were about 100 and 10 made on real ships but during mentioned experiments (about 100) made measurements just some parameters and made by different persons, who have not enough experience control such type of the ships. Complex measurements are limited to 11 because it was hard to gain higher number of qualified people who have experience to perform such type of ships maneuverers at port. In our future research work we will try to extend the number of measurements and involve more pilots. However, that process requires pilots to visit the laboratory available at Klaipeda University in order to preform ships’ operations on simulator and may face organisational challenges.
The separation of external factors was achieved in measurements by providing the settings to the simulator, the same as it was measured during the real ship movement at port (wind, wave, current, shallow water parameters). The settings in simulator included all mentioned external influences equal to the real ship sailing conditions. The appropriate comment was added to the main text.
Reviewer 2 Report
The manuscript Sustainability-806422 entitled "The method to decrease the emission from ships in ports areas" is very interesting. Therefore, even if the current version cannot be considered for the publication on the prestigious Sustainability journal I would like to suggest revision that requires a great effort to the authors.
Please, here below the comments that can help the authors to improve their manuscript.
- The abstract is missing about the current gap existent in the literature. It is relevant to justify the aims of research.
- The introduction need to improve. It is lacking about the references to the literature, the gap , the research questions, the structure of the manuscript. Besides, the authors mention the method but without clarifying.
- The manuscript is missing of theoretical background and a regulation framework for ships, obligations for shipowners, ports for the environmental issues. The authors have mentioned Di Vaio and Varriale study published on Sustainability in 2018, but they have not take the aspects more important, such as the systematization about the regulatory framework. The same authors have also published an other interesting Di Vaio, A., Varriale, L., & Alvino, F. (2018). Key performance indicators for developing environmentally sustainable and energy efficient ports: Evidence from Italy. Energy policy, 122, 229-240. In this article the authors developed a literature review including also all issues about the ships maneuvers and environmental effects. Finally, the authors can find other interesting articles on Maritime Policy & Management and Marine Policy, and so forth.
- The paper is missing of section related to the discussion and the comparision between findings of research and previous studies.
- The manuscript needs the section related to managerial and academic implications
- The results are weak
The current version of manuscript submit to the Sustainability Journal is more closer to a research letter then research paper. Thus, I suggest to the authors to review the manuscript but I think that you need almost two months (one for to study the literature and the regulation).
Good luck
Author Response
Suggestion 1: The abstract is missing about the current gap existent in the literature. It is relevant to justify the aims of research.
Respond: The gap was added to the article’s abstract. The sentence was added: “However, ship’s crew and ports pilots’ qualification influence on possibility to decrease the emission from ships during manoeuvring in ports areas has not been analyzed in detail so far. The article aims to develop a method to assess the possible decrease of the emission from ships in ports considering human factor influence.”.
Suggestion 2: The introduction need to improve. It is lacking about the references to the literature, the gap , the research questions, the structure of the manuscript. Besides, the authors mention the method but without clarifying.
Respond: The Introduction chapter was rewritten. The gap, the research questions and the structure of manuscript were added. The idea of proposed method was also clarified.
Suggestion 3: The manuscript is missing of theoretical background and a regulation framework for ships, obligations for shipowners, ports for the environmental issues. The authors have mentioned Di Vaio and Varriale study published on Sustainability in 2018, but they have not take the aspects more important, such as the systematization about the regulatory framework. The same authors have also published an other interesting Di Vaio, A., Varriale, L., & Alvino, F. (2018). Key performance indicators for developing environmentally sustainable and energy efficient ports: Evidence from Italy. Energy policy, 122, 229-240. In this article the authors developed a literature review including also all issues about the ships maneuvers and environmental effects. Finally, the authors can find other interesting articles on Maritime Policy & Management and Marine Policy, and so forth.
Respond: The second chapter was significantly rewritten and theoretical background was added. We generally described the regulation framework for ships, ports for the environmental issues, because article aimed to develop the method. New literature positions were added to reference list and analysed. That analysis allowed positioning the conducted research among other available research articles. The deep analysis of regulation framework for ships may form the topic of separate research article.
Suggestion 4: The paper is missing of section related to the discussion and the comparision between findings of research and previous studies.
Respond: The Section 5 with discussions elements was added to the article’s structure.
Suggestion 5: The manuscript needs the section related to managerial and academic implications
Respond: The paragraphs describing managerial and academic implications were placed at the discussion part of the article (Section 5). It was highlighted that appropriate teaching and training are very important for ships’ masters and pilots’ to gain high level of qualification, as well as may influence the volume of emission from ships.
Suggestion 6: The results are weak
Respond: In the case study the complex 11 measurements were carried out to prove the idea of the proposed method. The idea of the study was to show the way allowing to reduce the emission from ships. In our opinion the results show that proposed method is sufficient and may be introduced in practice. We agree that the number of measurements could be higher. It is limited to complex 11 (total experiments were about 100 and now mentioned in Article text) because it was hard to gain higher number of qualified people who have experience to perform this type of ships maneuverers at port. However, the research results are satisfactory and pay attention that level of pilots’ qualification may be different. In our future research we will try to extend the number of measurements and involve more qualified pilots, who will agree to take part in the research and provide ships maneuverers at the laboratory available at Klaipeda University and other Klaipeda similar laboratories.
Round 2
Reviewer 2 Report
Thank you for the efforts. The current version is improved and it can be considered for the publication